thermodynamics/statistical physics

Zigzag surface, asymmetrical Husimi lattice, phase transitions on surface

**Authors for correspondence:**
Ran Huang
e-mail: ranhuang@sjtu.edu.cn
Purushottam D. Gujrati
e-mail: pdg@uakron.edu

# Phase transitions of antiferromagnetic Ising spins on the zigzag surface of an asymmetrical Husimi lattice

## Ran Huang[1,2] and Purushottam D. Gujrati[3,4]

[1]State Key Laboratory of Microbial Metabolism and School of Life Sciences and Biotechnology, Shanghai Jiao Tong University, Shanghai 200240, People's Republic of China
[2]Department of Materials Technology and Engineering, Research Institute of Zhejiang University-Taizhou, Taizhou, Zhejiang 318000, People's Republic of China
[3]Department of Physics, and [4]Department of Polymer Science, The University of Akron, Akron, OH 44325, USA

RH, 0000-0001-7778-0920; PDG, 0000-0002-4092-5062

An asymmetrical two-dimensional Ising model with a zigzag surface, created by diagonally cutting a regular square lattice, has been developed to investigate the thermodynamics and phase transitions on surface by the methodology of recursive lattice, which we have previously applied to study polymers near a surface. The model retains the advantages of simple formulation and exact calculation of the conventional Bethe-like lattices. An antiferromagnetic Ising model is solved on the surface of this lattice to evaluate thermal properties such as free energy, energy density and entropy, from which we have successfully identified a first-order order–disorder transition other than the spontaneous magnetization, and a secondary transition on the supercooled state indicated by the Kauzmann paradox.

# 1. Introduction

The recursive lattice has been a classical methodology in statistical physics for several decades since its invention by Bethe [1] and Husimi [2]. With its impressive advantages of exact calculation and simple formulation due to the feature of recursive structure, many statistical or physical problems can seek an alternative but reliable solution from this modelling method, e.g. the travelling-salesman problem [3], the K-satisfiability [4], the glass transition [5] and so on. For most cases, especially when involving complex structures, it is impossible to solve arbitrary models on a square lattice so one is forced to find approximated solutions. Usually, one attempts to solve the model in a mean-field approximation, while

Gujrati [6] established that recursive lattice solutions are more reliable than the mean-field solutions, especially for the antiferromagnetic models.

On the other hand, the recursive feature of such lattices implies that the system is naturally homogeneous and suitable to describe bulky systems. Our group has already used the Bethe lattice to study inhomogeneous structures near a surface [7–9]. Here we extend to a one-/two-dimensional hybrid recursive lattice to study the phase transitions of Ising model on surface/interface, which is nonetheless still based on a symmetrical structure [10].

In this study, we have constructed an asymmetrical two-dimensional recursive lattice with a one-dimensional surface. By fractalizing the analogue of a diagonally cut regular square lattice, we can obtain a simple model with partial Husimi trees hung on a zigzag surface. Since this zigzag structure is infinite and homogeneous, it can be handled by recursive calculation [11,12], with approximating partial Husimi trees to be constant statistical contributions, which was derived from previous classical works, the exact calculation of this model appears to be feasible, like other Bethe-like models. An antiferromagnetic Ising model has been solved on the lattice, and we can locate the first-order phase transition and the Kauzmann paradox in the $\pm 1$ spin system.

# 2. Lattice and model

Conventionally, the surface of a two-dimensional square lattice is just the one-dimensional line confining the bulk with a lower coordination number of 3. Imagine we cleave the lattice along square diagonals, a zigzag surface can be generated for the one-half bulk, and the basic unit of this lattice is square in the bulk and triangle on the surface. The triangle unit is surrounded by two identical triangles and a bulk square, thus a straight connection of triangle units can be adopted to represent the surface. This triangle chain is infinite and has the recursive feature required for calculation. Considering that bulk parts are merely structures hung on the zigzag line, on each triangle we may attach an infinite partial Husimi tree to represent the bulk part. The scheme of fractalizing the halved structure to be a recursive lattice is shown in figure 1.

Since this zigzag surface recursive lattice (ZSRL) has coordination of four inside the bulk and alternating two or four (averagely three) on the surface, we hope that it is a good approximation to the regular square lattice with a diagonal boundary. The site labelling is shown in figure 2: the vertices on surface triangle are labelled $S$ and $S'$, where $S'$ marks the site closer to an imaginary origin point on the surface, following the direction of recursive calculation. Note the $S'$ in one triangle is $S$ in the unit of lower level while the origin is marked as level 0. The base site that links the surface triangle and bulk tree is labelled $S_B$.

Corresponding to the notation of neighbour interaction $J$, diagonal interaction $J_P$ and three spins interaction (triplet) $J'$, on the surface unit we label a bar on each term to differ them from bulk parameters. Then there are two neighbour interactions $\bar{J}$, one diagonal interaction $\bar{J}_P$ and one triplet ($\bar{J}'$) in the Hamiltonian of one triangle unit $\alpha$, which is

$$e_\alpha = -\bar{J}(S_B \cdot S + S_B \cdot S') - \bar{J}_P \cdot S \cdot S' - \bar{J}' \cdot S_B \cdot S \cdot S' - H(S + S_B), \tag{2.1}$$

where $H$ is the magnetic field applied to each spin. Note the $H$ of site $S'$ is not included in the last term, because it will be counted in the unit of next level. In equation (2.1), a negative value of $\bar{J}$ will raise the lowest energy with different neighbouring spins states, the system is thereafter the antiferromagnetic case, or vice versa. The detailed effects of energy parameters will be discussed in §4.2.

# 3. Calculation

## 3.1. Solutions on the surface

The general calculations of Ising model on Husimi square lattice can be found in previous works [6,13]. Here we follow the similar method of partial partition function (PPF) to achieve the solution $\bar{x}$, which presents the probability of one site to be occupied by + spin on the surface site. The triangle unit has $2^3 = 8$ possible configurations, four of which are with the base site $S' = +1$ and the others are with

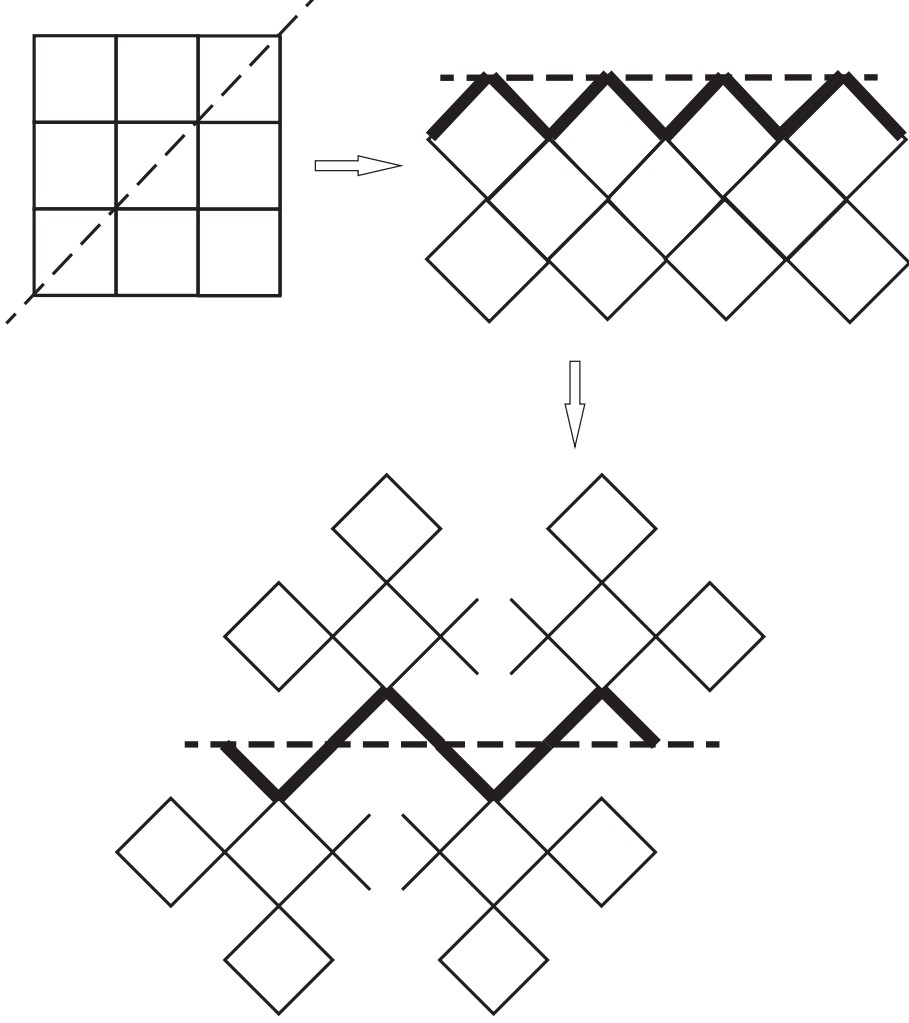

**Figure 1.** The diagonal cutting on a regular square lattice to obtain a zigzag surface, and the construction of recursive lattice on the zigzag surface. The thick line presents the surface edge.

$S' = -1$. We can derive two PPFs of the triangle on a level $m$ from PPFs of the higher level $m + 1$, the contribution from bulk $Z_B(S_B)$, and the local weight $w(\Gamma)$,

$$Z_m(+) = \sum_{\Gamma=1}^{4} Z_{m+1}(S_{m+1}) Z_B(S_B) w(\Gamma) \tag{3.1}$$

and

$$Z_m(-) = \sum_{\Gamma=5}^{4} Z_{m+1}(S_{m+1}) Z_B(S_B) w(\Gamma), \tag{3.2}$$

where $\Gamma$ is the index of configuration. Note that the total partition function (PF) of the entire system at the origin site $S_0$ is given by

$$Z_0 = Z_0(+)^2 \, e^{-\beta H} + Z_0(-)^2 \, e^{\beta H}. \tag{3.3}$$

This relationship is fundamental when we derive thermal quantities, e.g. free energy from the PPF.

We introduce ratios

$$\bar{x}_m = \frac{Z_m(+)}{Z_m(+) + Z_m(-)} \quad \text{and} \quad \bar{y}_m = \frac{Z_m(-)}{Z_m(+) + Z_m(-)} \tag{3.4}$$

as the solutions on surface at level $m$. By denoting

$$\bar{z}_m(S_m) = \begin{cases} \bar{x}_m & \text{if } S_m = +1 \\ \bar{y}_m & \text{if } S_m = -1, \end{cases} \tag{3.5}$$

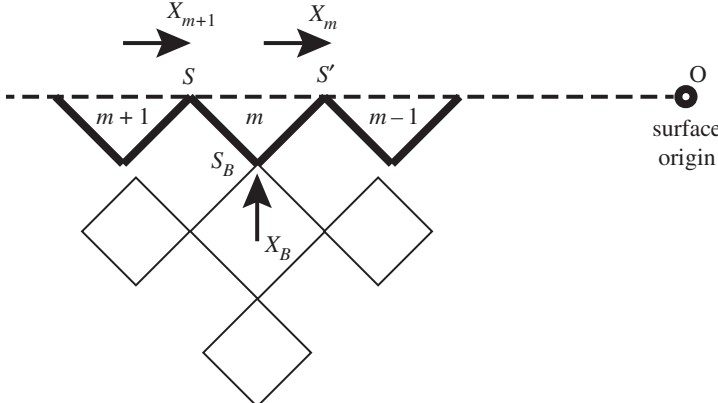

**Figure 2.** The site labelling and calculation scheme on ZSRL. Starting from a random point on the surface, the recursive approach is to proceed to an imaginary origin.

we have $Z_m(+) = B_m \bar{x}_m$ and $Z_m(-) = B_m \bar{y}_m$ with $B_m = Z_m(+) + Z_m(-)$, then with equations (3.1) and (3.2) we have

$$\bar{z}_m(S_m) = \frac{B_{m+1}B_B}{B_m} \sum_\Gamma \bar{z}_{m+1}(S_{m+1}) z_B(S_B) w(\Gamma). \tag{3.6}$$

By introducing a set of polynomials

$$\left. \begin{aligned} Q_{m+}(\bar{x}_{m+1}, x_B) &= \sum_{\Gamma=1}^4 \bar{z}_{m+1}(S_{m+1}) z_B(S_B) w(\Gamma), \\ Q_{m-}(\bar{y}_{m+1}, y_B) &= \sum_{\Gamma=5}^4 \bar{z}_{m+1}(S_{m+1}) z_B(S_B) w(\Gamma) \\ Q_m(\bar{x}_{m+1}, x_B) &= Q_{m+}(\bar{x}_{m+1}, x_B) + Q_{m-}(\bar{y}_{m+1}, y_B) = \frac{B_m}{B_{m+1}B_B}, \end{aligned} \right\} \tag{3.7}$$

and

we can derive the solution $\bar{x}_m$ at the $m$th level as a function of the ratio $\bar{x}_{m+1}$ on higher level and the bulk solution $x_B$,

$$\bar{x}_m = \frac{Q_{m+}(\bar{x}_{m+1}, x_B)}{Q_m(\bar{x}_{m+1}, x_B)}. \tag{3.8}$$

Taking $x_B$ as the solution on a joint site between surface triangle and bulk square, imagine we start the recursive calculation from a position deep in the bulk, then approach the thermodynamic contributions to the surface, thereby $x_B$ is simply the fix-point solution of Husimi lattice. In this way, we can count the contribution of the infinite bulk tree as a constant input and focus on the recursive approach of $\bar{x}$ along the surface. It should be addressed that this set-up of constant $x_B$ ignores the possible backward effect from surface to the bulk, which may bias the numerical value of $x_B$. However, this approximation is acceptable to make the model simple and solvable.

With a constant $x_B$, by equation (3.8) we can recursively calculate the solution on surface for a number of iterations until we reach a fix-point solution. The form of equation (3.8) implies that, regardless of the system being antiferro- or ferromagnetic, a uniform 1-cycle solution is expected on the surface, while antiferromagnetic Ising model presents an alternating 2-cycle solution as the ordered state [13]. A negative neighbour interaction $\bar{J}$ prefers to anti-align the $S$ versus $S_B$ and $S'$ versus $S_B$ pair. Unless we set the diagonal interaction $\bar{J}_P$ also to be negative and large enough to outweigh $\bar{J}$, the system will prefer the same spin states on $S$ and $S'$.

Recall that bulk solutions can be either a 1-cycle solution to present the metastable state, or a set of 2-cycle solutions as the ordered state [13]. Taking the 1-cycle $x_B$, which is usually 0.5 with $H = 0$, we will obtain a fixed $\bar{x} = 0.5$ also corresponding to a metastable surface; for the 2-cycle solutions, we can substitute either fixed solution as $x_B$, and it will affect the calculation on $\bar{x}$ to bias the surface fix-point solution to 0 or 1. For example, due to antiferromagnetism, at $T = 0$ with $H = 0$, we will have 0 and 1 solutions in the bulk, then if we take $x_B = 0$, the surface solution will be captured as $\bar{x} = 1$, and vice versa. Our results confirmed that the thermodynamics calculated based on either selection are identical.

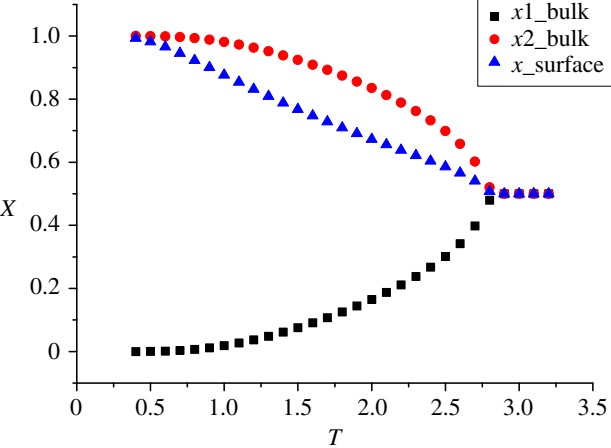

**Figure 3.** Solution on the surface and its comparison to the 2-cycle bulk solutions.

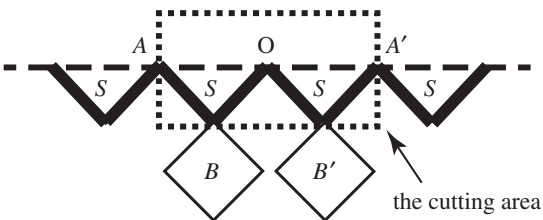

**Figure 4.** The cutting scheme around the surface origin $O$ for free energy calculation.

Figure 3 shows the ordered solution on the surface with comparison of 2-cycle bulk solutions, the energy terms are set as $J = -1, \bar{J} = -1$ and others to be 0. The metastable 0.5 solution is not presented in figure 3. For convenience, we are still going to call the stable solution and corresponding thermodynamics '2-cycle' in the following discussion, although both stable and metastable $\bar{x}$ are actually in 1-cycle form for ZSRL.

It can be observed that at high $T$ all the solutions are 0.5, that spins anywhere have an equal probability to be $\pm 1$. Below the Curie point, the spins undergoes self-magnetization (even without an external field $H$), and the neighbouring spins prefer the $+/-$ alternating arrangement, which is the lowest energy state. The value of 2-cycle $x_B$ of bulk Ising model is referred from previous reports [10,13]. Along with the bulk solutions, the spins on surface also present preference on unitary direction under the bulk Curie point; however, we will show that the actual phase transition occurs far below the $T_C$ (bulk).

One more thing should be addressed here. We know from exact solutions that for the square lattice the self-magnetization occurs at $T_C = 2/\log\left(1 + \sqrt{2}\right) \sim 2.27$. In the recursive lattice (RL), the bulk transition temperature is 2.8 as shown in figure 3, which is not all that close to the known $T_C$, but closer than the results of mean-field theory. We believe this overestimate is acceptable. The aspect that the RL method provides results between the 'real' exact solution and the mean-field theory seems to be generic. Similar results had also been observed in other works, e.g. the three-dimensional cube lattice [13]. However, the nature of this overestimate has not been investigated yet.

## 3.2. Free energy calculation

We follow the Gujrati trick to calculate the free energy of a local area by recursive approach [6,13]. The scheme will be briefly described here, as shown in figure 4. Owing to the uniform structure and solution on the surface, we can randomly select a site as the origin point $O$. The local area is chosen to be two triangle units joint on the origin. Imagine we cut off two sub-trees contributing to the point $A$ and $A'$ then rejoin them together to make an identical but smaller ZSRL, and hook up two partial bulk trees hung on $B$ and $B'$ together to make a full Husimi square lattice; therefore we have $F_{\text{total}} = F_{\text{local}} +$

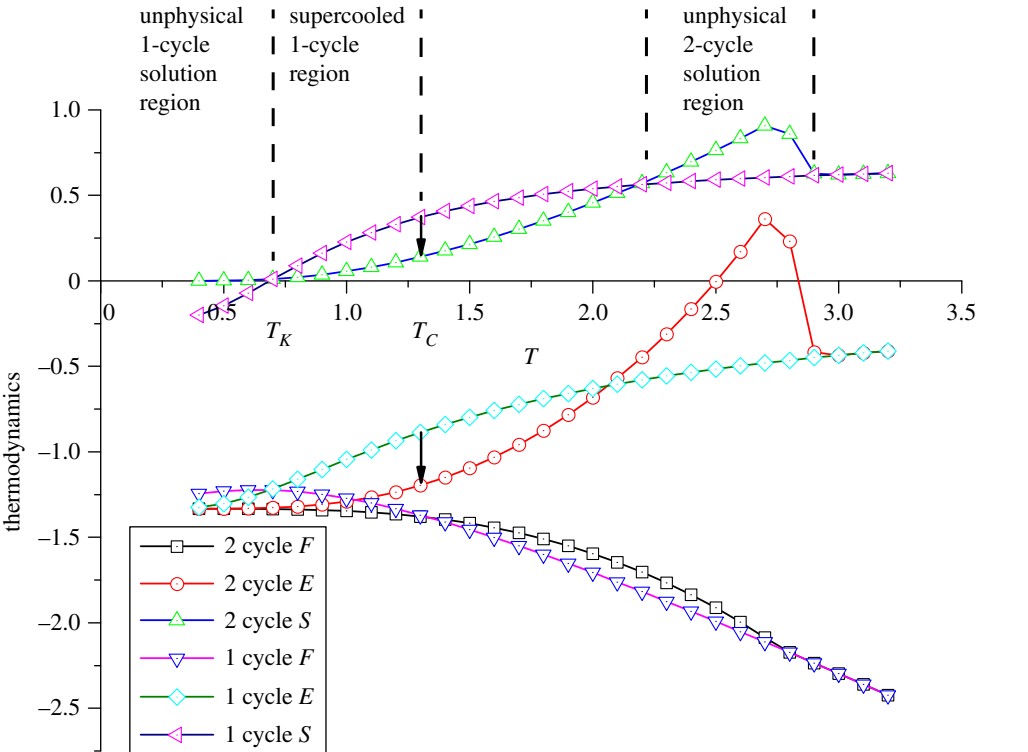

**Figure 5.** The thermodynamics behaviours on the surface with $J = -1, \bar{J} = -1$ and other parameters to be 0.

$F_{\text{bulk}} + F_{\text{smaller}}$. With the definition of Helmholtz free energy $F = -k_B\, T \log Z$, the $F$ per site in the local area is given as

$$F_{\text{site}} = -\frac{1}{3}T\log\left(\frac{Z_0}{Z_1 \cdot Z_B}\right), \tag{3.9}$$

since there are three full sites in the local area (on whole site $S_0$ and four half-shared sites $S_1$, $S_B$) and $k_B$ is normalized to be 1.

Recalling equations (3.1)–(3.3), it is easy to break down the local free energy into a function of PPFs. Then from the relations between $Z(\pm)$, $Q$ and $x$ derived in the previous section, we can calculate the above function as

$$F_{\text{site}} = -\frac{1}{3}T\log\left(\frac{Q_0{}^2}{x_B{}^2\, e^{\beta H} + (1 - x_B)^2\, e^{-\beta H}}\right), \tag{3.10}$$

then the entropy and energy (per site) can be easily achieved by $S = -\mathrm{d}F/\mathrm{d}T$ and $E = F + TS$.

# 4. Results and discussion

## 4.1. The thermodynamics and transitions on the surface

The thermal behaviours of the reference set-up with $J = -1, \bar{J} = -1$ and other parameters to be 0 is shown in figure 5. The free energy of two solutions differ at $T = 2.8$. Usually this bifurcation indicates the spontaneous magnetization (Curie point) in normal spin models. However, here on the surface the magnetization does not bend the free energy of alternating spins arrangement below the disordered 0.5 solution but upward, which is unphysical with a sharply increased entropy. Therefore, the 2-cycle solution at this point is only a numerical existence and the system must follow the curve of 1-cycle solution, until a cross point is reached at $T = 1.33$, where the system makes a transition from the amorphous to the crystalline ordering, i.e. the order–disorder transition at critical temperature $T_C$. Unlike the conventional self-magnetization where entropy is continuous, the entropy here must undergo a discontinuous jump like a first-order transition. In this way, we may conclude that the critical order–disorder transition on the surface is not the Curie point, but much lower than that of the bulk. This can be understood as that, due to the asymmetry and

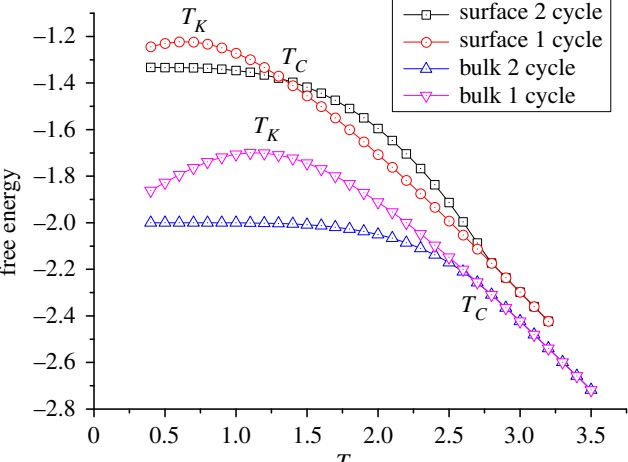

**Figure 6.** The free energy comparison of Husimi bulk system and ZSRL with $J = -1, \bar{J} = -1$ and other parameters to be 0.

smaller coordination numbers on the surface, spins on the outer layer are less dragged by the bulk portions. Below the real $T_C$ (bulk) and the apparent Curie point on the surface, the spins on surface can maintain their 'melted' state in a deep temperature region, and this does not fall into the concept of supercooled liquid, since the corresponding crystal state in particular temperature region is unfeasible. Intuitively, this phenomenon may easily refer to the analogue of ice premelting [14]; nonetheless, this work only focuses on the preliminary modelling and we are not going to further expand this point.

For the 1-cycle solution below $T_C$, the system can undergo cooling without any phase transition and shares features of a supercooled state. With further cooling process, the entropy of 2-cycle solution approaches zero, while the entropy of 1-cycle becomes negative at $T = 0.69$, which is the Kauzmann paradox at $T_K$ [13].

The result indicates that a free surface dramatically decreases the transition temperatures. Figure 6 shows the free energy comparison of Husimi bulk system and ZSRL. This observation agrees with others' work on phase transitions on the surface or thin film, for example, the glass transition of polymer system in confined geometry [7–9,15]. The fact that similar reduction can be observed in our monoatomic model implies that the lower transition temperature on surface/thin film basically originates from the dimension reduction and less interaction constraints.

## 4.2. The effects of secondary energy parameters

Besides mathematical curiosity, there is always a primary expectation to describe and study real systems for establishing a theoretical model. In this way, other than the interaction $J$ and $\bar{J}$ between the nearest neighbours, further interactions are included to make the model more versatile to describe various systems, as shown in equation (2.1). With adjustable combinations of energy parameters set-up, we can manipulate the thermal behaviour of system and the transition temperatures to better match the reality in particular situations. In other words, more adjustable parameters may serve as useful tools for the theoretical modelling to be correlated with experimental parameters. By the control of $J = -1$ and other parameters to be 0 inside the bulk, we explored the effects of $\bar{J}$, $\bar{J}_P$ and $\bar{J}'$. The secondary parameters are expected to either comply or compete with $J$, and some interesting phase behaviours are found with particular set-ups.

### 4.2.1. The surface nearest-neighbour interaction $\bar{J}$

Considering the feature of asymmetry on the boundary, we set the nearest-neighbour interaction on the surface, denoted as $\bar{J}$, to be differentiated and adjustable from the $J$, to make the model capable to describe some particular situations, e.g. the surface tension. The transition temperatures with $J = -1$, other parameters to be 0, and $\bar{J} = -0.5, -0.7, -0.9, -1.1, -1.3$ and $-1.5$ are shown in table 1.

As negative $\bar{J}$ complies with antiferromagnetic set-up $J = -1$, the larger absolute value of $\bar{J}$ makes the system more stable and increases both critical and ideal glass transition temperature; and the relative length of supercooled region, indicated by the ratio $T_C/T_K$, gradually increases. However, for the $\bar{J} = -0.5$ decreased from $-0.7$, both $T_C$ and $T_K$ fall while the latter changes more dramatically,

**Table 1.** The transition temperature variations with different $\bar{J}$.

| $\bar{J}$ | $T_C$ | $T_K$ | $T_C/T_K$ |
|---|---|---|---|
| −0.5 | 0.85 | 0.40 | 2.13 |
| −0.7 | 1.10 | 0.60 | 1.83 |
| −0.9 | 1.20 | 0.63 | 1.90 |
| −1 | 1.33 | 0.69 | 1.93 |
| −1.1 | 1.50 | 0.77 | 1.95 |
| −1.3 | 1.80 | 0.90 | 2.00 |
| −1.5 | 2.10 | 1.01 | 2.08 |

**Table 2.** The transition temperature variations with different $\bar{J}_P$.

| $\bar{J}_P$ | $T_C$ | $T_K$ | $T_C/T_K$ |
|---|---|---|---|
| −0.4 | 0.85 | 0.40 | 2.13 |
| −0.2 | 1.10 | 0.60 | 1.83 |
| 0 | 1.33 | 0.69 | 1.93 |
| 0.2 | 1.54 | 0.75 | 2.05 |
| 0.4 | 1.73 | 0.85 | 2.04 |
| 0.6 | 1.90 | 0.92 | 2.07 |

**Table 3.** The transition temperature variations with different $\bar{J}'$.

| $\bar{J}'$ | $T_C$ | $T_K$ | $T_C/T_K$ |
|---|---|---|---|
| −0.3 | 1.90 | 0.76 | 2.50 |
| −0.1 | 1.55 | 0.70 | 2.21 |
| 0 | 1.33 | 0.69 | 1.93 |
| 0.1 | 1.05 | 0.69 | 1.52 |

raising a much larger supercooled region. This critical phenomenon observes that a loosened surface with too weak interactions is easier to be supercooled.

### 4.2.2. The diagonal interaction $\bar{J}_P$

The diagonal interaction $\bar{J}_P$ between the two top sites in the triangle unit is the only competition to the nearest-neighbour $\bar{J}$. Transition temperatures with $\bar{J}_P = \pm 0.2$, $\pm 0.4$ and 0.6 are summarized in table 2. As expected, same polarity of $\bar{J}_P$ and $\bar{J}$ obstructs the ordering degree with reduced $T_C$ and $T_K$, while positive $\bar{J}_P$ complies with $\bar{J}$ and gives higher $T$s and larger supercooled region. A similar phenomenon of sharply reduced $T$s and enlarged $T_C/T_K$ ratio is also observed with $\bar{J}_P = -0.4$, which implies that a vigorous competition between $\bar{J}_P$ and $\bar{J}$ has the same effect of weak $\bar{J}$ to make the surface preferring supercooled state.

### 4.2.3. The triplet interaction $\bar{J}'$

To avoid analysing the complex three body interactions, the triplet interaction term $\bar{J}'$ is introduced to count a compacted polarity of triangle unit, which may act similarly to the magnetic field $H$. Table 3 summarizes the $T$s with $\bar{J}' = -0.3$, $-0.1$ and 0.1. It is found that negative $\bar{J}'$ increases the transition temperatures or vice versa, and the thermodynamics is very sensitive to $\bar{J}'$, i.e. a slight variation will

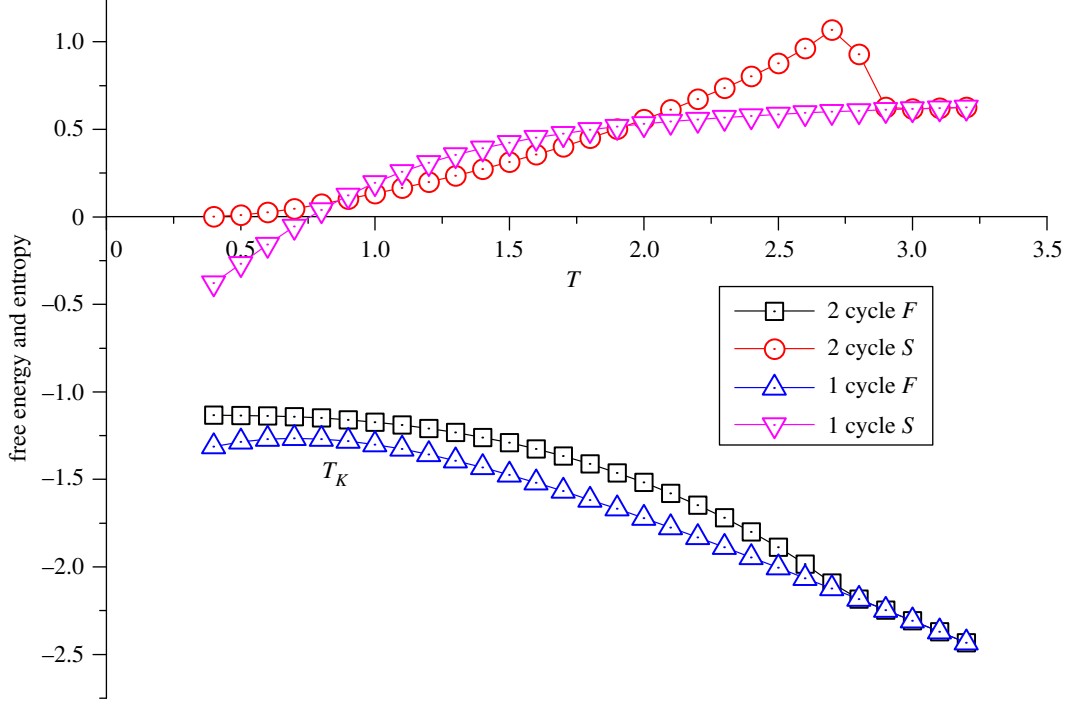

**Figure 7.** A special case of ZSRL with $\bar{J} = -1$ and $J' = 0.3$.

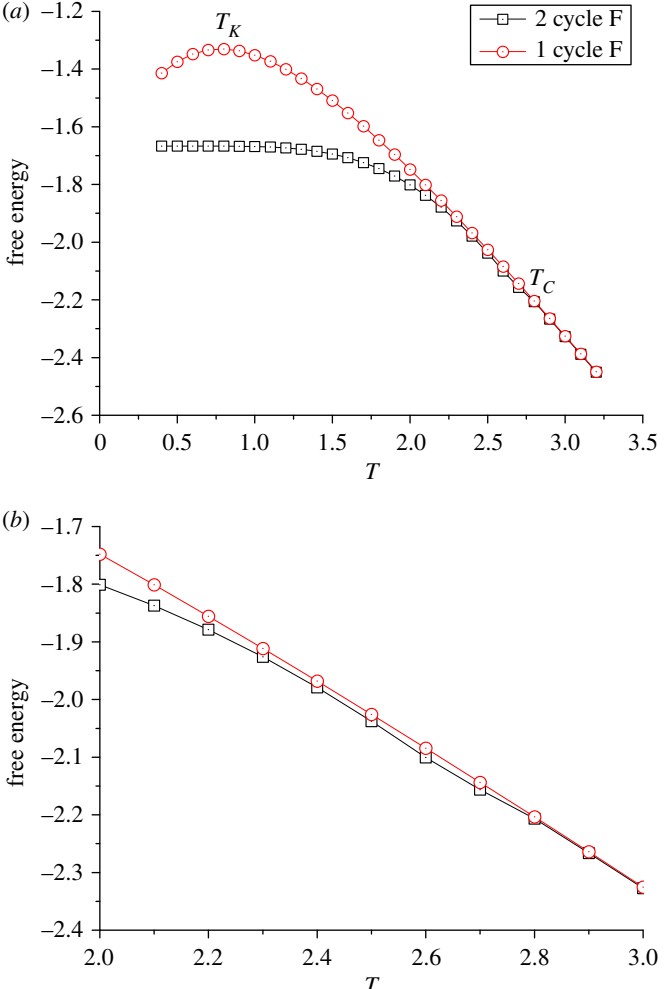

**Figure 8.** A special case of ZSRL with $\bar{J} = -1$ and $J' = -0.5$: (a) the free energy of 1- and 2-cycle solutions; (b) the enlargement of free energy around $T = 2.8$.

dramatically change the overall thermal behaviours. The set-up with absolute value of $\bar{J}'$ larger than 0.3 can converge the system to some bizarre states, which will be detailed in the following section.

## 4.3. Two special cases with various $\bar{J}'$

Since the $\bar{J}'$ plays a dominant role in ZSRL, the value of $\bar{J}'$ is limited to be relatively small. Abnormal behaviours can be observed with $\bar{J}' = 0.3$ and $\bar{J}' = -0.5$. In the first case, as shown in figure 7, the 2-cycle solution will never have a lower free energy than 1-cycle. Even it has a lower entropy at low temperature, the order–disorder transition cannot be located since there is no cross point. On the other hand, the 1-cycle solution still undergoes Kauzmann paradox. Therefore, the only reasonable understanding is that, under this condition, regardless of the thermal state in the bulk, crystal state is not achievable on the surface. With temperature decreasing, we will only have supercooled liquid and the subsequent glassy state.

The system goes to another extremity for $\bar{J}' = -0.5$: figure 8 shows that, below the free energy differentiating point, the 2-cycle solution consistently becomes more stable than 1-cycle, like the normal behaviour of regular antiferromagnetic Ising models, and thereby the order–disorder transition becomes spontaneous magnetization; below $T_C$ the system can either be in crystal ordering or in the metastable supercooled state.

## 5. Conclusion

A zigzag surface recursive lattice has been constructed to describe a regular square lattice with one-dimensional boundary. The zigzag structure is taken as a surface assembled by triangle units, and halved Husimi trees are hung on the triangle units to represent the bulk portions. With the coordination number of four inside the bulk and average three on the surface, this model is considered to be a good approximation to a regular square lattice with surface.

The antiferromagnetic Ising model is solved on the lattice, with the constant $x_B$ retrieved from regular Husimi lattice to count the bulk contribution, and a uniform solution is obtained on the surface to represent the ordered state. Then the thermodynamics of local area around the origin on surface can be derived by conventional techniques from $\bar{x}$ and $x_B$. The transition temperatures are found to be dramatically reduced on the surface compared with in the bulk, and this reduction is simply due to the dimension downgrade and less interaction constrains on the surface.

The effects of various interaction energy parameters other than nearest-neighbour $\bar{J}$ are investigated. These interactions could either increase or decrease the stability of system and change the transition temperatures according to the Hamiltonian. In addition to the effect of parameters, we have found several interesting behaviours with particular energy set-up.

Data accessibility. The code and data are available from the Dryad Digital Repository: https://doi.org/10.5061/dryad.99t5k4s [16].

Authors' contributions. R.H. designed the model, did the programming, calculation and analysis, and wrote the manuscript. P.D.G. directed the research, derived the theoretical formulation and calculation method, edited the manuscript and approved its submission.

Competing interests. We declare no competing interests.

Funding. This work is financially supported by the National Natural Science Foundation of China (11505110), the China Postdoctoral Science Foundation (2016M591666) and the Taizhou Municipal Science and Technology Program (1701gy15 and 1801gy16).

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
