## [Reviewer comments · Royal Society Open Science]

Review History

RSOS-181500.R0 (Original submission)

Review form: Reviewer 1

Is the manuscript scientifically sound in its present form?

Yes

Are the interpretations and conclusions justified by the results?

Yes

Is the language acceptable?

Yes

Is it clear how to access all supporting data?

Yes

Do you have any ethical concerns with this paper?

No

Have you any concerns about statistical analyses in this paper?

No

Recommendation?

Accept as is

Comments to the Author(s)

In this paper, the authors study a zigzag surface recursive lattice with a regular square lattice. The zigzag structure is taken as a surface assembled by triangle units, and halved Husimi trees are hung on the triangle units to represent the bulk portions. With the coordination number of 4 inside the bulk and average 3 on the surface, this model is considered to be a good approximation to a regular square lattice with surface. The model is considered in the case of antiferromagnetic interaction on the lattice and a uniform solution is obtained on the surface to represent the ordered state. The thermodynamical properties of the model are also investigated and the transition temperatures are found to be dramatically reduced on the surface comparing which in the bulk. The effects of several interaction parameters on the system are investigated and these interactions could either increase or decrease the stability of system and change the transition temperatures according to the Hamiltonian.

The overall impression of the paper is rather positive. The subject and method are quite clear, the calculations are correct. Thus the paper can be published in its present form.

Review form: Reviewer 2

Is the manuscript scientifically sound in its present form?

No

Are the interpretations and conclusions justified by the results?

No

Is the language acceptable?

No

Is it clear how to access all supporting data?

Yes

Do you have any ethical concerns with this paper?

No

Have you any concerns about statistical analyses in this paper?

No

Recommendation?

Major revision is needed (please make suggestions in comments)

Comments to the Author(s)

I believe this paper should undergo major revisions before being properly reviewed as I have several issues.

I should stress from the beginning these issues may be fundamental or they may be from misunderstanding what is being discussed.

Neither case is optimal, but of course the latter may be fixed by addressing these misunderstandings in the text.

The first issue concerns the usefulness of studying the recursive lattice (RL) in the context of the regular square lattice (SL).

There does not seem to be any real argument or evidence that the RL is a reasonable approximation to the SL, beyond mention of coordination numbers matching (and even then, only on average on the surface).

Indeed, figure 3 would argue it is a bad approximation as $x1_{\text{bulk}}$ and $x2_{\text{bulk}}$ become equal well beyond the transition temperature for the SL Ising model (~ 2.27).

Your removal of any connectivity (through the bulk) between each surface triangle does not seem conducive to getting physical results; loops and connectivity are vital component of spin systems on lattices.

In the SL there is one bulk which, if long range ordered, applies an effective field to the surface forcing surface order.

Do you force each limb of the RL to apply the same effective field to each part of the surface, despite the decoupling?

Clear physical statements need to be made about how the 1 bulk to many bulks approximation impacts the problem.

The second is about the interpretation of your data.

I will focus on figure 5 to concreteness.

You argue (and I agree) that the 2-cycle solution cannot be physical above $T = 1.33$ and so the 1-cycle solution describes the system.

However, I do not see any real evidence that it is not also just a numerical artefact below this temperature and hence it crossing the 1-cycle solution does not necessarily indicate a phase transition.

To clarify, I'm not saying it doesn't indicate a phase transition, only that I do not see evidence that the 2-cycle goes from physically absurd to describing the system.

Third, and most easily remedied, is that I have some small mathematical issues.

In equation (6) I don't understand where the power of two comes from.

It appears all that is going on is to take equations (2) and (3) and set $Z_i = B_i \bar{z}_i$, which would give the pre-factor as $B_{m+1} B_B / B_m$.

I'm happy to be wrong, but some explanation it is not what I think should appear in the paper if so.

Also, I feel equation (12) would benefit from some additional explanation.

Finally, section IVB does not have any indication as to why someone would be interested in adding and varying these parameters.

It seems like just tables of data with little justification.

I am not saying it is necessarily uninteresting or that it should not be included, just that it must be justified.

There are also several writing based problems which hinder understanding.

While I would be happy to give a comprehensive list, I think it is inappropriate to do so before the above issues are resolved and this document may be reviewed properly.

Decision letter (RSOS-181500.R0)

06-Nov-2018

Dear Dr Huang,

The editors assigned to your paper ("Phase transitions of antiferromagnetic Ising spins on the zigzag surface of an asymmetrical Husimi lattice") have now received comments from reviewers. We would like you to revise your paper in accordance with the referee and Associate Editor suggestions which can be found below (not including confidential reports to the Editor). Please note this decision does not guarantee eventual acceptance.

Please submit a copy of your revised paper before 29-Nov-2018. Please note that the revision deadline will expire at 00.00am on this date. If we do not hear from you within this time then it will be assumed that the paper has been withdrawn. In exceptional circumstances, extensions may be possible if agreed with the Editorial Office in advance. We do not allow multiple rounds of revision so we urge you to make every effort to fully address all of the comments at this stage. If deemed necessary by the Editors, your manuscript will be sent back to one or more of the original reviewers for assessment. If the original reviewers are not available, we may invite new reviewers.

- Data accessibility

If you wish to submit your supporting data or code to Dryad (<http://datadryad.org/>), or modify your current submission to dryad, please use the following link:
<http://datadryad.org/submit?journalID=RSOS&manu=RSOS-181500>

- **Competing interests**

- **Authors' contributions**

- **Acknowledgements**

- **Funding statement**

Please note that Royal Society Open Science charge article processing charges for all new submissions that are accepted for publication. Charges will also apply to papers transferred to Royal Society Open Science from other Royal Society Publishing journals, as well as papers submitted as part of our collaboration with the Royal Society of Chemistry (<http://rsos.royalsocietypublishing.org/chemistry>). If your manuscript is newly submitted and subsequently accepted for publication, you will be asked to pay the article processing charge, unless you request a waiver and this is approved by Royal Society Publishing. You can find out more about the charges at <http://rsos.royalsocietypublishing.org/page/charges>. Should you have any queries, please contact openscience@royalsociety.org.

Kind regards,

Royal Society Open Science Editorial Office
Royal Society Open Science
openscience@royalsociety.org

on behalf of Dr Robert Young (Associate Editor) and Prof. Miles Padgett (Subject Editor)

Associate Editor's comments (Dr Robert Young):

Please see the detailed comments from the reviews of this work. I'm concerned that the underlying physics in the paper is flawed and that it may never be suitable for publication. If you are able to address these concerns then please make major revisions to your work and resubmit the manuscript.

Comments to Author:

Reviewers' Comments to Author:

Reviewer: 1

Comments to the Author(s)

In this paper, the authors study a zigzag surface recursive lattice with a regular square lattice. The zigzag structure is taken as a surface assembled by triangle units, and halved Husimi trees are hung on the triangle units to represent the bulk portions. With the coordination number of 4 inside the bulk and average 3 on the surface, this model is considered to be a good approximation to a regular square lattice with surface. The model is considered in the case of antiferromagnetic interaction on the lattice and a uniform solution is obtained on the surface to represent the ordered state. The thermodynamical properties of the model are also investigated and the transition temperatures are found to be dramatically reduced on the surface comparing which in the bulk. The effects of several interaction parameters on the system are investigated and these interactions could either increase or decrease the stability of system and change the transition temperatures according to the Hamiltonian.

The overall impression of the paper is rather positive. The subject and method are quite clear, the calculations are correct. Thus the paper can be published in its present form.

Reviewer: 2

Comments to the Author(s)

I believe this paper should undergo major revisions before being properly reviewed as I have several issues.

I should stress from the beginning these issues may be fundamental or they may be from misunderstanding what is being discussed.

Neither case is optimal, but of course the latter may be fixed by addressing these misunderstandings in the text.

The first issue concerns the usefulness of studying the recursive lattice (RL) in the context of the regular square lattice (SL).

There does not seem to be any real argument or evidence that the RL is a reasonable approximation to the SL, beyond mention of coordination numbers matching (and even then, only on average on the surface).

Indeed, figure 3 would argue it is a bad approximation as $x1_{bulk}$ and $x2_{bulk}$ become equal well beyond the transition temperature for the SL Ising model (~ 2.27).

Your removal of any connectivity (through the bulk) between each surface triangle does not seem conducive to getting physical results; loops and connectivity are vital component of spin systems on lattices.

In the SL there is one bulk which, if long range ordered, applies an effective field to the surface forcing surface order.

Do you force each limb of the RL to apply the same effective field to each part of the surface, despite the decoupling?

Clear physical statements need to be made about how the 1 bulk to many bulks approximation impacts the problem.

The second is about the interpretation of your data.

I will focus on figure 5 to concreteness.

You argue (and I agree) that the 2-cycle solution cannot be physical above $T = 1.33$ and so the 1-cycle solution describes the system.

However, I do not see any real evidence that it is not also just a numerical artefact below this temperature and hence it crossing the 1-cycle solution does not necessarily indicate a phase transition.

To clarify, I'm not saying it doesn't indicate a phase transition, only that I do not see evidence that the 2-cycle goes from physically absurd to describing the system.

Third, and most easily remedied, is that I have some small mathematical issues.

In equation (6) I don't understand where the power of two comes from.

It appears all that is going on is to take equations (2) and (3) and set $Z_i = B_i \bar{z}_i$, which would give the pre-factor as $B_{m+1} B_B / B_m$.

I'm happy to be wrong, but some explanation it is not what I think should appear in the paper if so.

Also, I feel equation (12) would benefit from some additional explanation.

Finally, section IVB does not have any indication as to why someone would be interested in adding and varying these parameters.

It seems like just tables of data with little justification.

I am not saying it is necessarily uninteresting or that it should not be included, just that it must be justified.

There are also several writing based problems which hinder understanding.

While I would be happy to give a comprehensive list, I think it is inappropriate to do so before the above issues are resolved and this document may be reviewed properly.

Author's Response to Decision Letter for (RSOS-181500.R0)

See Appendix A.

RSOS-181500.R1 (Revision)

Review form: Reviewer 2

Is the manuscript scientifically sound in its present form?

Yes

Are the interpretations and conclusions justified by the results?

Yes

Is the language acceptable?

Yes

Is it clear how to access all supporting data?

Yes

Do you have any ethical concerns with this paper?

No

Have you any concerns about statistical analyses in this paper?

No

Recommendation?

Accept with minor revision (please list in comments)

Comments to the Author(s)

Overall, I believe that while this paper needs some modifications, they are not as prohibitive as previously thought.

As such I would recommend that it is accepted pending those changes.

I believe my initial objections are answered sufficiently by your likening of your approach to mean-field theory.

That is, you believe that solving the recursive lattice problem provides the same sort of information about the real square lattice model that mean-field theory would, but also that it would do so with greater accuracy.

However, it would be useful for the reader to have this stated clearly very early on.

Otherwise they may worry, as I did, about the applicability of exact statements on the recursive lattice to the square lattice.

My second objection, concerning figure 3, remains.

I was perhaps not entirely clear - I was not talking about any Monte Carlo-esque methods but rather exact results.

I view the two bulk calculations as being analogous to the magnetisation in the ferromagnetic square lattice Ising model, albeit scaled and shifted.

I completely agree with what you say in your response: at zero temperature you would expect perfect antiferromagnetism, and hence $x1_bulk = 1$, $x2_bulk = 0$; at low temperature these values would be perturbed only slightly; at high temperature the spins are completely disordered and so neighbours are equally likely to be aligned or anti-aligned, hence $x1_bulk = x2_bulk = 0.5$.

My query is about at what temperature the phase transition between order and disorder occurs. We know from exact solutions that for the square lattice this occurs at $T_c = 2/\log(1+\sqrt{2}) \sim 2.27$.

My objection was that figure 3 implies a transition temperature of 2.75, which is not all that close to the known T_c .

You may argue that 2.75 is closer than what mean-field theory would predict, but this should be stated for the reader.

Additionally, I wonder if anything can be said about why it is an overestimate; is this generic for all results obtained in this manner?

Finally, I have attached a list of writing based suggestions you may wish to consider (see Appendix B). This list is not comprehensive but will hopefully provide a useful starting point.

Decision letter (RSOS-181500.R1)

15-Jan-2019

Dear Dr Huang:

On behalf of the Editors, I am pleased to inform you that your Manuscript RSOS-181500.R1 entitled "Phase transitions of antiferromagnetic Ising spins on the zigzag surface of an asymmetrical Husimi lattice" has been accepted for publication in Royal Society Open Science subject to minor revision in accordance with the referee suggestions. Please find the referees' comments at the end of this email.

The reviewers and Subject Editor have recommended publication, but also suggest some minor revisions to your manuscript. Therefore, I invite you to respond to the comments and revise your manuscript.

- Ethics statement

- Data accessibility

If you wish to submit your supporting data or code to Dryad (<http://datadryad.org/>), or modify your current submission to dryad, please use the following link:
<http://datadryad.org/submit?journalID=RSOS&manu=RSOS-181500.R1>

- Competing interests

- Authors' contributions

- Acknowledgements

- Funding statement

Because the schedule for publication is very tight, it is a condition of publication that you submit the revised version of your manuscript before 24-Jan-2019. Please note that the revision deadline will expire at 00.00am on this date. If you do not think you will be able to meet this date please let me know immediately.

Supplementary files will be published alongside the paper on the journal website and posted on

the online figshare repository (<https://figshare.com>). The heading and legend provided for each supplementary file during the submission process will be used to create the figshare page, so please ensure these are accurate and informative so that your files can be found in searches. Files on figshare will be made available approximately one week before the accompanying article so that the supplementary material can be attributed a unique DOI.

on behalf of Dr Robert Young (Associate Editor) and Professor Miles Padgett (Subject Editor)
 openscience@royalsociety.org

Reviewer comments to Author:
 Reviewer: 2

Comments to the Author(s)

Overall, I believe that while this paper needs some modifications, they are not as prohibitive as previously thought.

As such I would recommend that it is accepted pending those changes.

I believe my initial objections are answered sufficiently by your likening of your approach to mean-field theory.

That is, you believe that solving the recursive lattice problem provides the same sort of information about the real square lattice model that mean-field theory would, but also that it would do so with greater accuracy.

However, it would be useful for the reader to have this stated clearly very early on.

Otherwise they may worry, as I did, about the applicability of exact statements on the recursive lattice to the square lattice.

My second objection, concerning figure 3, remains.

I was perhaps not entirely clear - I was not talking about any Monte Carlo-esque methods but rather exact results.

I view the two bulk calculations as being analogous to the magnetisation in the ferromagnetic square lattice Ising model, albeit scaled and shifted.

I completely agree with what you say in your response: at zero temperature you would expect perfect antiferromagnetism, and hence $x1_bulk = 1$, $x2_bulk = 0$; at low temperature these values would be perturbed only slightly; at high temperature the spins are completely disordered and so neighbours are equally likely to be aligned or anti-aligned, hence $x1_bulk = x2_bulk = 0.5$.

My query is about at what temperature the phase transition between order and disorder occurs. We know from exact solutions that for the square lattice this occurs at $T_c = 2/\log(1+\sqrt{2}) \sim 2.27$.

My objection was that figure 3 implies a transition temperature of 2.75, which is not all that close to the known T_c .

You may argue that 2.75 is closer than what mean-field theory would predict, but this should be stated for the reader.

Additionally, I wonder if anything can be said about why it is an overestimate; is this generic for all results obtained in this manner?

Finally, I have attached a list of writing based suggestions you may wish to consider. This list is not comprehensive but will hopefully provide a useful starting point.

Author's Response to Decision Letter for (RSOS-181500.R1)

See Appendix C.

Decision letter (RSOS-181500.R2)

29-Jan-2019

Dear Dr Huang,

I am pleased to inform you that your manuscript entitled "Phase transitions of antiferromagnetic Ising spins on the zigzag surface of an asymmetrical Husimi lattice" is now accepted for publication in Royal Society Open Science.

on behalf of Dr Robert Young (Associate Editor) and Miles Padgett (Subject Editor)
openscience@royalsociety.org

Appendix A

Thanks the referee's careful review and positive feedback. The original comments are pasted below as *italic*, followed by our response item by item. Along with the response, the revisions we have made in the manuscript are marked as red.

Reviewer: 2

Comments to the Author(s)

I believe this paper should undergo major revisions before being properly reviewed as I have several issues.

I should stress from the beginning these issues may be fundamental or they may be from misunderstanding what is being discussed.

Neither case is optimal, but of course the latter may be fixed by addressing these misunderstandings in the text.

The first issue concerns the usefulness of studying the recursive lattice (RL) in the context of the regular square lattice (SL).

There does not seem to be any real argument or evidence that the RL

is a reasonable approximation to the SL, beyond mention of coordination numbers matching (and even then, only on average on the surface).

R: For most cases, especially when involving complex structures, It is not possible to solve arbitrary models on a square lattice so one is forced to find approximated solutions. Usually, one attempts to solve the model in a mean-field approximation. Ref. 9 establishes that recursive lattice solutions are more reliable than the mean-field solutions, especially for antiferromagnetic models. Indeed, recursive lattice solutions have become the state-of-the art in statistical mechanics.

The average coordination 3 is an approximation of zigzag surface itself, instead of RL. Considering a regular 2D square lattice being diagonally cut, a zigzag outlayer will reveal, like the second part of fig1 (Plus we modified the fig1 a bit to make the zigzag on regular cleavage lattice clearer).

Indeed, figure 3 would argue it is a bad approximation as $x1_{bulk}$ and $x2_{bulk}$ become equal well beyond the transition temperature for the SL Ising model (~2.27).

R: There might be some misunderstanding on Fig3. Because this is the content of bulk Ising model, which is discussed in other previous reports (Ref. 6 and 10 for example), so we did not talk too much on this. But “ $x1_{bulk}$ and $x2_{bulk}$ become equal well” makes perfect sense. Since it is the antiferromagnetic system, below the Curie point (T_c), the spins will undergo self-magnetization (even without an external field H), and prefer the $+ - + - \dots$ alternating arrangement, which is the lowest energy state. Therefore, the probability of $+$ and $-$ spins must be “equally well”.

Furthermore, the referee rising this confusion might because of that, in other classical simulation works, such as Monte Carlo or MD, an external field is necessary to induce the magnetization below T_c , otherwise it has a chance to achieve a temporary metastable, unpolarized state (which is the 1-cycle solution in our model). And the external field would bias the $x1_{bulk}$ and $x2_{bulk}$ to be asymmetrical.

Actually, this is another small advantage of our calculation method, to observe the self-magnetization without external field

We have added some discussions on Fig3, especially on the x_{bulk} , to clarify similar confusions.

Your removal of any connectivity (through the bulk) between each surface triangle does not seem conducive to getting physical results; loops and connectivity are vital component of spin systems on lattices.

The referee is correct that connectivity is vital for correlations. This is the price we pay for using the recursive approximation. However, we account for the bulky influence by introducing the back field x_{B} in our computation. The idea is to find a solution that is physically consistent but not exact numerically. It is this idea of “physically consistent” solution that is used to draw conclusions when discussing Fig. 5. It is this point that has bothered the referee.

Although the connections between bulk trees are eliminated, since

each single bulky tree is infinite and has an independent circumstance inside, the thermodynamic contributions of various structures such as loops and so on can be counted inside the tree. The philosophy here is very much like the mean-field method, since the complex interaction between components are so messed up and uncountable, we deduce the effects to be a single factor on each site itself, as a “mean field”.

In the SL there is one bulk which, if long range ordered, applies an effective field to the surface forcing surface order.

Do you force each limb of the RL to apply the same effective field to each part of the surface, despite the decoupling?

Clear physical statements need to be made about how the 1 bulk to many bulks approximation impacts the problem.

Like explained above, It is right that we do normalize the effect from bulk to be a uniform factor, x_B , (very much like a “field”), We consider this to be a reasonable treatment. However, under current framework we cannot verify this or to quantify how bad it could be. But we may do some Indirect comparisons with some other lattice models and even

experimental works on the similar problems. For example, another RL for interface we had developed (Ref. 6), the MC work of polymer glass transition on thin film (P. Doruker and W. L. Mattice, *Macromolecules* 1998, 31, 1418-1426), or MD works on similar systems (J.A Torres, P. F. Nealey, and J. J. de Pablo, *Phys Rev Lett* 2000, 85, 15). Our results are found to be quite comparable to theirs, as we all observed the reduction of transition T , and even the reduction ratios are quite close to the experimental results (J.A. Forrest, K. Dalnoki-Veress, J.R. Dutcher, *Phys. Rev. E* 56. 1997. 5705.). However, this part is not very relevant to the properties of the model itself, so we did not mention these comparisons in the manuscript.

The second is about the interpretation of your data.

I will focus on figure 5 to concreteness.

You argue (and I agree) that the 2-cycle solution cannot be physical above $T = 1.33$ and so the 1-cycle solution describes the system.

However, I do not see any real evidence that it is not also just a numerical artefact below this temperature and hence it crossing the 1-cycle solution does not necessarily indicate a phase transition.

To clarify, I'm not saying it doesn't indicate a phase transition, only that I do not see evidence that the 2-cycle goes from physically absurd to describing the system.

R: It is not clear why the referee thinks that the solution below $T=1.33$ is numerically suspect. The recursive numerical calculation is exact; we always make sure that we have come to a fixed point, where the numerical error is physically insignificant. The crossing of the free energy is well beyond this physically insignificant "error." It is the free energy that determines which solution is thermodynamically relevant. There is nothing wrong with a solution being unphysical in one region, and only to become a physical one in another region.

Third, and most easily remedied, is that I have some small mathematical issues.

In equation (6) I don't understand where the power of two comes from.

It appears all that is going on is to take equations (2) and (3) and set $Z_i = B_i \bar{z}_i$, which would give the pre-factor as $B_{m+1} B_B / B_m$.

I'm happy to be wrong, but some explanation it is not what I think should appear in the paper if so.

Also, I feel equation (12) would benefit from some additional explanation.

R: The referee is correct. With a careful check on our original notes, here it is typo that the actual form should be $B_{m+1}B_B / B_m$, we have corrected this. And we added some detailed discussions on eq12 to help audience to understand the “from and go” of eq12, and how the thermals are calculated from PPFs..

Finally, section IVB does not have any indication as to why someone would be interested in adding and varying these parameters.

It seems like just tables of data with little justification.

I am not saying it is necessarily uninteresting or that it should not be included, just that it must be justified.

R: We have talked more on the motivation and meanings of more adjustable parameters in the section IVB. Besides the interests in

mathematical puzzling and guessing, we always expect the theoretical model to be able to mimic and study real systems. In this way, we include the interactions of further range to make the model versatile to describe various systems. With adjustable combinations of energy parameters setup, we can manipulate the thermal behavior of system and the transition temperatures, to better match the reality in particular situation. In another word, theoretical modelings always face the challenge to be correlated with experimental parameters, more adjustable parameters may serve as useful tools for the purpose.

There are also several writing based problems which hinder understanding.

While I would be happy to give a comprehensive list, I think it is inappropriate to do so before the above issues are resolved and this document may be reviewed properly.

R: We have reviewed the manuscript and polished the writings at several points. Again, thanks the referee's precious and helpful review and we hope we have fulfilled all the concerns.

Appendix B

Page/line	Current text	Edited text
1/14	with a zigzag surface by diagonally cutting a regular square lattice has been developed	with a zigzag surface, created by diagonally cutting a regular square lattice, has been developed
1/17	The model retains advantages	The model retains the advantages
1/18	Antiferromagnetic Ising model	An antiferromagnetic Ising model
1/34	intensively interested	intensively studied
1/41	Antiferromagnetic Ising model have been solved	An antiferromagnetic Ising model have been solved
1/51	to represent the surface, then this triangle chain	to represent the surface. This triangle chain
2/42	is shown in fig.2: The vertices	is shown in fig.2: the vertices
2/53	where H is the magnetic field of a single spin.	where H is the magnetic field applied to each spin.
2/54	In eqn.1, the negative value	In eqn.1, a negative value
4/4	comma after equation (7)	fullstop after equation (7)
4/27	from a position far deep in the bulk, then approach the thermodynamic	from a position deep in the bulk then approach the thermodynamic
4/28	the contribution of infinite bulk tree as a constant input, and	the contribution of the infinite bulk tree as a constant input and
4/29	It should be addressed that, this setup	It should be addressed that this setup
4/31	value of x_B , however this approximation	value of x_B . However this approximation
4/33	regardless of antiferro- or ferromagnetic system	regardless of the system being antiferro- or ferromagnetic
4/39	ordered state [10], taking the 1-cycle	ordered state [10]. Taking the 1-cycle
6/31	differ at $T = 2.8$, usually	differ at $T = 2.8$. Usually
6/32	However here on the surface, the magnetization	However, here on the surface the magnetization
6/34	increased entropy, therefore the 2-cycle	increased entropy. Therefore the 2-cycle
6/34	the system must undergoes following the curve	the system must follow the curve
6/37	And unlike the conventional	Unlike the conventional
6/37	must undergoes a discontinuous jump	must undergo a discontinuous jump
6/39	but much lower than which is in the bulk	but much lower than that of the bulk
9/30	to count the bulk contribution, a uniform solution is obtained	to count the bulk contribution, and a uniform solution is obtained

Appendix C

We sincerely appreciate the referee's 2nd round review and very helpful corrections. We have corrected the mistakes pointed, and reviewed the manuscript again for language polish.

For the remaining concerns, we have also made some revisions. The main change is some added introduction on the advantages of RL in section I, to remind readers that recursive lattice and mean-field theory are both approximation to the real lattice, while RL is believed to be with greater accuracy in some situations.

And we added a discussion on the overestimate of RL method comparing to exact results, but smaller than mean-field, which is the last paragraph in section III.A. The referee is correct that this is a necessary point to be addressed to readers, and the overestimate seems to be generic, which were also observed in other works. However, we have not studied this nature yet.

Thanks again.